# Heat stress illness outcomes and annual indices of outdoor heat at U.S. Army installations

**Stephen A. Lewandowski**[1,2]*, **Marianthi-Anna Kioumourtzoglou**[1], **Jeffrey L. Shaman**[1]

**1** Department of Environmental Health Sciences, Columbia University Mailman School of Public Health, New York, New York, United States of America, **2** Department of Preventive Medicine and Biostatistics, Uniformed Services University of the Health Sciences, Bethesda, Maryland, United States of America

* stephen.lewandowski@usuhs.edu

**Data Availability Statement:** Climate index data are available from https://doi.org/10.5281/zenodo.5903145. Outcome data are available from the Defense Medical Epidemiology Database (DMED) at https://www.health.mil/Military-Health-Topics/

## Abstract

This study characterized associations between annually scaled thermal indices and annual heat stress illness (HSI) morbidity outcomes, including heat stroke and heat exhaustion, among active-duty soldiers at ten Continental U.S. (CONUS) Army installations from 1991 to 2018. We fit negative binomial models for 3 types of HSI morbidity outcomes and annual indices for temperature, heat index, and wet-bulb globe temperature (WBGT), adjusting for installation-level effects and long-term trends in the negative binomial regression models using block-bootstrap resampling. Ambulatory (out-patient) and reportable event HSI outcomes displayed predominately positive association patterns with the assessed annual indices of heat, whereas hospitalization associations were mostly null. For example, a one-degree Fahrenheit (˚F) (or 0.55˚C) increase in mean temperature between May and September was associated with a 1.16 (95% confidence interval [CI]: 1.11, 1.29) times greater rate of ambulatory encounters. The annual-scaled rate ratios and their uncertainties may be applied to climate projections for a wide range of thermal indices to estimate future military and civilian HSI burdens and impacts to medical resources.

## Introduction

Heat stress illnesses (HSIs) pose a preventable, potentially fatal, health threat with serious impacts to military training and readiness [1, 2]. HSIs occur when the effects of environmental heat stress, combined with metabolic heat generated from physical activity, exceed thermoregulatory and heat exchange capacities, resulting in elevated core temperature [3]. This heat strain manifests as a continuum of outcomes, including heat stroke, heat exhaustion, edema, cramps, and fainting. In the U.S. Army, diagnosed cases of heat stroke and heat exhaustion have increased in recent years as average annual temperatures and high temperature records continue to rise [4, 5]. Military service members who train in the Continental U.S. (CONUS) experience elevated risks from heat exposure compared to similar age groups in the general population due to increased time outdoors, high physical exertion levels, clothing burden, and equipment loads. Types of demanding physical tasks vary by military occupational specialty;

Combat-Support/Armed-Forces-Health-Surveillance-Division/Data-Management-and-Technical-Support/Defense-Medical-Epidemiology-Database. DMED is available to authorized users such as U.S. military medical providers, epidemiologists, medical researchers, safety officers or medical operations/ clinical support staff for surveying health conditions in the U.S. military. Civilian collaborators in military medical research and operations may also have access to DMED with documentation supporting their arrangements.

**Funding:** S.L. was supported by the US Army Long Term Health Education and Training program and the National Institutes of Health/National Institute of Environmental Health Sciences training grant T32 ES007322. J.S. was supported by a gift from the Morris-Singer Foundation. J.S. and Columbia University disclose partial ownership of SK Analytics. J.S. also reports receiving consulting fees from BNI. All other authors declare no competing interests. The funders had no role in study design, data collection and analysis, decision to publish, or preparation of the manuscript.

**Competing interests:** The authors have declared that no competing interests exist.

however, common motions consist of lifting and carrying or lifting and lowering, and activities include foot marches, physical training, obstacle courses, and combat training lanes [6, 7]. Military heat stress exposures may be broadly generalizable to civilian populations with a similar age distribution, exposure to outdoor conditions, and exertional levels, including athletic and occupational settings [8]. However, specific prevention guidelines should account for risk factor differences between groups [9].

The environmental properties affecting heat exchange include air temperature, air humidity, wind speed, and solar, sky, and ground radiation [3]. A wide range of methods and indices exist to classify the thermal environment as it relates to thermal stress and physiological effects [10]. The primary index used by the U.S. Army is the wet bulb globe temperature (WBGT). The WBGT is a weighted average of natural wet-bulb temperature (weight, $w = 70\%$), globe thermometer temperature ($w = 20\%$), and dry-bulb temperature ($w = 10\%$) in outdoor, non-shaded conditions [11]. Another commonly reported metric is the U.S. National Weather Service's (NWS) heat index. The NWS heat index (HI) represents an apparent temperature measure of thermal comfort based on air temperature and relative humidity and serves as a basis for excessive heat warnings [12]. The U.S. Army Public Health Center also applies the NWS HI as an indicator for heat risk days, defined as days with an HI greater than 90˚F (32.2˚C) for more than one hour [4]. Although WBGT and HI are most often applied to short-term (hourly, daily, or heat wave event) exposures, averages or aggregates from these instruments can also assist with characterization of long-term (seasonal, annual) heat and humidity risks. The relationship between daily-scale indices and HSI encounters at military sites was assessed in a separate study [13].

The objective of this study was to characterize the association between indices of heat and annual HSI morbidity outcomes among active-duty soldiers at ten CONUS Army installations in the context of rising temperature and humidity conditions. The resulting estimates can be used to quantify projected climate change impacts and inform long-term planning assumptions with implications for military and civilian populations.

## Materials and methods

### Health outcome data

We obtained HSI outcome counts and rates of hospitalization (in-patient), ambulatory visits (out-patient), and reportable medical events from the Defense Medical Epidemiology Database (DMED), which contains summarized, non-Privacy Act data for active component service members from the Defense Medical Surveillance System (DMSS) [14]. Hospitalization and ambulatory data include encounters from Department of Defense (DoD) and non-DoD treatment facilities. Reportable events are defined in the Armed Forces Reportable Medical Events Guidelines and Case Definitions and represent conditions that pose a significant threat to public health and military operations [15]. The DMED application is accessible through the Armed Forces Health Surveillance Division at https://www.health.mil/dmed/ for authorized users and validated medical researchers [16]. We queried primary diagnosis International Classification of Diseases (ICD) codes for active-duty U.S. Army servicemembers. For ICD-9, used through 2015, we applied 992-series codes, categorized as "effects of heat and light" [17]. We used ICD-10 series T67 codes for 2016–2018 data [18]. The counts and rates in this study aggregate all conditions within these code groups, with heat stroke and heat exhaustion representing the majority of cases across each of the three outcome types. The rates are based on active component servicemember populations at each location for each year. Hospitalization data were available from 1990–2018, ambulatory from 1997–2018, and reportable events from 1995–2018. We excluded the initial years for hospitalizations and ambulatory encounters

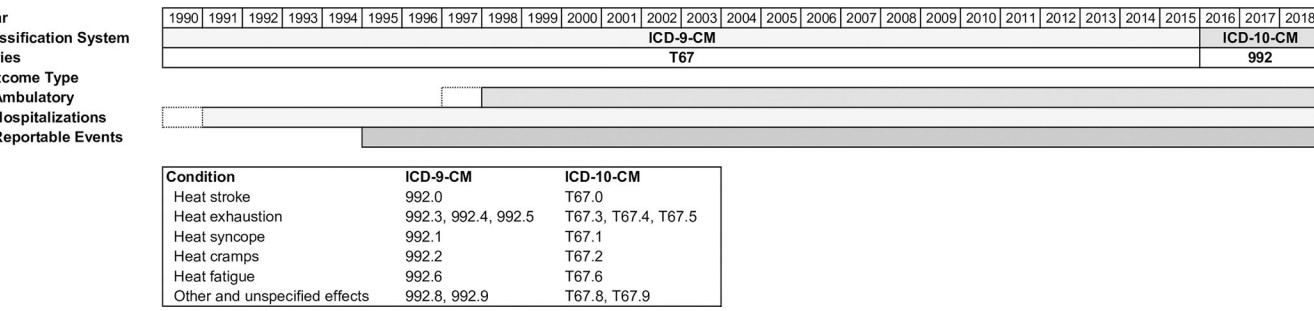

**Fig 1. Clinical classification codes and included outcome types by year.** The shaded bars depict the included years for each outcome type.

(1990 and 1997, respectively) from analyses due to indicators of incomplete reporting. Fig 1 displays the included outcome types by year and lists the clinical classification codes. Additionally, we queried injuries and illnesses of all types to consider potential long-term trends due to changes in reporting systems or access-to-care and to assess the relative burden of disease due to HSIs. We selected ten U.S. Army CONUS installations based on previously reported *Medical Surveillance Monthly Report (MSMR)* HSI rates and exploratory DMED findings [1]. The ten included locations (listed in Table 2) account for over 78% of all CONUS active-duty Army HSI cases for the examined years. The first excluded location, Fort Irwin, CA, reported less than half the HSI cases as the tenth ranked site, Fort Bliss, TX.

## Meteorology data

Meteorological estimates from the North American Land Data Assimilation System 2 (NLDAS-2) forcing dataset served as the primary source of weather and atmospheric data [19]. NLDAS is a National Aeronautics and Space Administration (NASA) / National Oceanic and Atmospheric Administration (NOAA)-led multi-institution project that constructs gridded surface meteorological datasets through the assimilation and merging of fields derived from gauge-based and remotely-sensed observations and re-analyses, with validation from ground-based observations [19]. Its land surface model integrates atmospheric observations from sources including meteorological stations, radiosondes, and satellites to derive land surface states [20]. NLDAS-2 data are available on a 1/8th-degree spatial scale at hourly frequencies from 1979 to present. We selected NLDAS grid cells containing the centroid of each installation based on shapefiles from the Department of Defense (DoD) Military Installations, Ranges, and Training Areas (MIRTA) Dataset [21].

NLDAS fields include air temperature at 2 meters above the surface, specific humidity at 2 meters above the surface, surface pressure, wind speed, and bias-corrected surface downward shortwave radiation. We calculated relative humidity from specific humidity, temperature, and atmospheric pressure; heat index (HI) from temperature and relative humidity based on a US National Weather Service algorithm [12]; and outdoor WBGT from air temperature, relative humidity, solar irradiance, barometric pressure, and wind speed using the method of Liljegren *et al.*, based on principles of heat and mass transfer [22, 23].

We compiled annual indices of heat through multiple aggregations of hourly temperature, HI, and WBGT estimates in absolute and relative terms, averaged either over the entire calendar year or the heat season, defined as 01 May through 30 September. We included full-year averages, as a prior study assessed that approximately 17% of all HSI cases occurred during non-summer months, variable by location [24]. Absolute measures included annual mean temperatures and counts of heat risk days or hours above specified thresholds based on heat

**Table 1. Classification of included annual indices.**

| Comparison | Averaging Period | Index Type | Annual Indices |
|---|---|---|---|
| Absolute | Full Year | Temperature | Mean of daily mean; mean of daily maximum; hours > 90˚F (32.2˚C); hours > 100˚F (37.8˚C) |
| | | Heat Index | Mean of daily mean; mean of daily maximum; hours > 90˚F (32.2˚C); hours > 105˚F (40.6˚C) |
| | | WBGT | Mean of daily mean; mean of daily maximum; hours > 85˚F (29.4˚C); hours > 90˚F (32.2˚C) |
| | Heat Season | Temperature | Mean of daily mean; mean of daily maximum |
| | | Heat Index | Mean of daily mean; mean of daily maximum |
| | | WBGT | Mean of daily mean; mean of daily maximum |
| Relative (1990–2019 baseline) | Full Year | Temperature | Mean of daily mean anomaly; days > 1 standard deviation of baseline |
| | | Heat Index | Mean of daily mean anomaly; days > 1 standard deviation of baseline |
| | | WBGT | Mean of daily mean anomaly; days > 1 standard deviation of baseline |
| | Heat Season | Temperature | Mean of daily mean anomaly; days > 1 standard deviation of baseline |
| | | Heat Index | Mean of daily mean anomaly; days > 1 standard deviation of baseline |
| | | WBGT | Mean of daily mean anomaly; days > 1 standard deviation of baseline |

category cut-offs for HI and WBGT. Mean values were calculated over 24-hour periods to capture minimum temperatures, which can impact recovery from heat exposure. We calculated relative measures with reference to 1990 to 2019 climatologies for each day of the year and each location. These relative indices included annual mean daily anomalies and counts of days one standard deviation above daily temperature climate norms for mean daily values. Table 1 summarizes the index classifications.

## Statistical analyses

To evaluate time trends in our exposure metrics, we fit linear models regressing each index of heat on time for each installation. We evaluated outcome measures in a similar manner, with simple linear regressions for rates of each outcome type over time, by installation and with combined rates ($\frac{sum\ of\ counts}{sum\ of\ population}$) for all ten installations.

We applied negative binomial regression to model the over-dispersed count outcomes for hospitalizations, ambulatory encounters, and reportable events [25]. The index of heat served as the exposure of interest, in increments of ˚F, number of days, or number of hours. Fahrenheit was selected as a base unit because this scale is commonly used by the U.S. military and is primarily featured in heat categorization and prevention tables. We set indicator variables for each installation to account for potentially confounding factors varying across installations and set the active-duty Army population of each installation for each year as an offset. Our resulting regression formula for the log of the rate predicted by the exposure and installation indicator variables is: $\log\left(\frac{outcome\ count_i}{population_i}\right) = \hat{\beta}_0 + \hat{\beta}_1 index\ value + \sum_{j=2}^{10} \hat{\beta}_j I\ (installation_i = j)$, with Fort Bliss, TX set as the reference installation by virtue of it having the lowest HSI encounter counts among the included sites. We accounted for confounding by year, which is associated with long-term trends in both the exposure and outcome, by applying a block bootstrap approach that shuffles replicated selections of the data to reduce effects of serial correlation [26]. The time variable includes elements which we are limited in our ability to decompose, such as changes in access to care, admission protocols, coding practices, and reporting systems in addition to soldier demographics, fitness levels, and training intensities. We hypothesize that if we were to only include *year* as a term in a standard model without a blocked bootstrap approach, the trend would capture a portion of the outcome variability associated with the changes in heat we are investigating and bias estimates towards the null, while failure to adjust for trends through time in any manner would bias results away from the null.

To construct block bootstraps, we randomly selected two-year intervals with replacement and assembled these intervals into a new series with the approximate length of the base time series. We conducted 2,000 replications of this process for each model, calculated beta coefficients for each iteration, and constructed nonparametric basic (empirical) bootstrap confidence intervals [27, 28]. We assessed sensitivity by comparing non-bootstrap models (with and without a year term), original single observation bootstraps, and three-year block interval bootstraps. In the two-year block models, we also examined bias-corrected and accelerated (BCa) bootstrap intervals, which incorporate parameters for the proportion of bootstrap estimates less than the observed statistic and for the skewness of the bootstrap distribution [29]. We conducted all statistical and spatial analyses using R Statistical Software (version 3.6.1) [30]. The R code is available at https://github.com/sal2222/annual_heat.

## Results

We found that CONUS active-duty Army HSI ambulatory and reportable event rates increased over the study period. Hospitalizations also increased, but the rate did not reach statistical significance at α = 0.05. Assessing outcome patterns for all types of injuries and illnesses, we observed that ambulatory rates sharply increased over time and hospitalization rates generally declined from 1991 to 1997 and then steadied. Reportable event rates displayed random variability but were the most stable outcome measure over time. The mean HSI counts and rates for each installation are listed in Table 2, along with mean burden, representing the percent of all encounters or events attributed to HSIs. Ambulatory events were most reported, with a mean total of 2,081 per year over the assessed period for the included sites. Reportable events averaged 394 per year and hospitalizations averaged 109 per year. Fourteen installation-outcome type pairs exhibited a positive, linear trend for annual rate at α = 0.05 over the included years and two had a negative trend. Fig 2 displays the positive trends of the combined HSI rates from the ten installations over time (p < 0.001 for ambulatory and reportable event regression slopes, p = 0.12 for hospitalizations). The overall active component population

**Table 2. Heat stress illness outcomes (all HSI types).**

| Installation | Ambulatory (1998–2018) | | | Hospitalization (1991–2018) | | | Reportable Events (1995–2018) | | |
|---|---|---|---|---|---|---|---|---|---|
| | Mean Count (SD) | Mean Rate (SD) | Mean Burden % (SD) | Mean Count (SD) | Mean Rate (SD) | Mean Burden % (SD) | Mean Count (SD) | Mean Rate (SD) | Mean Burden % (SD) |
| Fort Bliss, TX | 28.33 (16.6) | 1.57 (0.58) | 0.01 (0.00) | 1.75 (1.88) | 0.11 (0.12) | 0.10 (0.09) | 3.50 (3.88) | 0.22 (0.23) | 0.63 (0.68) |
| Fort Benning, GA | 535.48 (290.58) | 26.51 (15.00) | 0.15 (0.05) | 38.00 (20.14) | 1.93 (0.96) | 2.52 (1.52) | 67.38 (54.45) | 3.42 (2.84) | 18.69 (11.43) |
| Fort Bragg, NC | 702.52 (271.78) | 15.51 (5.28) | 0.13 (0.05) | 31.00 (13.01) | 0.72 (0.31) | 1.04 (0.53) | 140.83 (60.03) | 3.21 (1.41) | 11.57 (5.51) |
| Fort Campbell, KY | 191.76 (112.63) | 6.81 (4.08) | 0.05 (0.02) | 10.00 (5.48) | 0.38 (0.19) | 0.54 (0.31) | 45.08 (37.7) | 1.59 (1.35) | 6.77 (6.04) |
| Fort Hood, TX | 110.81 (36.94) | 2.68 (1.03) | 0.02 (0.01) | 7.71 (4.09) | 0.19 (0.11) | 0.24 (0.15) | 27.46 (25.12) | 0.64 (0.56) | 1.33 (0.90) |
| Fort Jackson, SC | 265.29 (202.53) | 27.84 (22.31) | 0.13 (0.09) | 3.25 (2.88) | 0.34 (0.32) | 0.38 (0.38) | 52.92 (83.18) | 5.63 (9.09) | 13.22 (17.25) |
| Fort Leonard Wood, MO | 59.86 (51.3) | 6.24 (5.50) | 0.03 (0.02) | 3.21 (2.56) | 0.36 (0.29) | 0.39 (0.38) | 7.00 (5.79) | 0.76 (0.65) | 3.70 (4.12) |
| Fort Polk, LA | 74.67 (49.06) | 9.21 (6.24) | 0.06 (0.03) | 4.64 (3.01) | 0.53 (0.38) | 0.72 (0.63) | 22.00 (23.8) | 2.77 (3.09) | 8.35 (7.98) |
| Fort Riley, KS | 42.67 (27.28) | 2.89 (1.54) | 0.02 (0.01) | 1.71 (1.65) | 0.12 (0.11) | 0.17 (0.16) | 8.96 (5.74) | 0.67 (0.42) | 2.60 (1.94) |
| Fort Stewart, GA | 69.57 (40.98) | 4.22 (2.44) | 0.03 (0.01) | 7.86 (15.67) | 0.58 (1.38) | 0.57 (0.93) | 18.71 (17.68) | 1.14 (1.08) | 2.88 (2.12) |

Rates are per 1,000 persons per year. Burden is calculated as the percentage of HSI encounters compared to the total of all documented injuries and illnesses. Light gold shaded cells indicate a positive linear regression slope for HSI rate over time at α = 0.05. Light blue shaded cells indicate a negative linear regression slope for HSI rate over year at α = 0.05 (hospitalization rates at Fort Bliss and Fort Stewart).

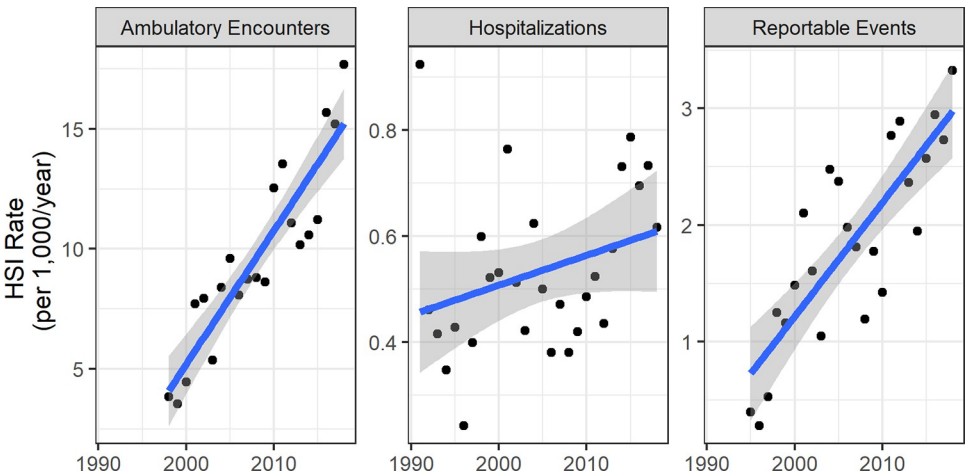

**Fig 2. Combined HSI outcome rates for ten CONUS Army installations.** The line represents a linear model and the shaded area models 95% confidence levels. Note that the scales vary by outcome category by orders of magnitude.

from the 10 included installations varied over time. The total population increased between 1991–2011 and decreased between 2011–2018, ranging from 176,490 in 1991 to 249,915 in 2011.

We also detected positive long term (decadal) trends among indices of heat, compiled over the entire calendar year or restricted to heat season months, across the assessed CONUS Army installations. Table 3 displays summary statistics for mean annual indices and highlights indices with significant positive linear time trends at α = 0.05. Each installation had at least one index reflecting a warming trend over a 27-year period. Annual climate trends are plotted in Fig 3 for mean and maximum index averages and means of location-specific anomalies.

Regression model analysis of annual indices and HSI outcomes found positive association patterns (rate ratio; RR > 1 at α = 0.05) for ambulatory encounters and reportable events (Fig 4, Table 4). Hospitalizations displayed a pattern of null associations with annual indices. HI and WBGT-based indices were more likely to indicate a stronger positive association than ambient temperature indices. Quantifying a sample result, we found that a 1˚F (0.55˚C) increase in mean temperature between May and September is associated with a 1.16 (95% CI: 1.11, 1.29) times greater rate of ambulatory encounters among active-duty Army soldiers at CONUS locations, controlling for installation-specific effects. Indices averaged over the full calendar year were more likely to display stronger positive associations than those averaged over heat season months. Indices based on hourly counts above threshold values for HI and WBGT trended towards the null. Relative indices (mean anomalies, days 1 standard deviation above a climate normal) reflected comparable association patterns to their counterpart absolute indices.

In our sensitivity analyses of various models, non-bootstrap negative binomial models adjusted for year returned RR estimates closer to the null than 2-year block bootstrap models. Results from standard bootstrap models (single year replacement) approximated negative binomial models without adjustment for year. Models resampled in 3-year blocks were more likely to return CIs spanning the null than 2-year block models.

## Discussion

Few, if any, prior studies have calculated an association between heat exposure at an annual scale and HSI morbidity response among a physically active population in the United States.

**Table 3. Summary of absolute, mean degree-based annual indices of heat (1991–2018).**

| | Full Year | | | Heat Season (May—September) | | |
|---|---|---|---|---|---|---|
| | Temperature | Heat Index | WBGT | Temperature | Heat Index | WBGT |
| | Mean (SD) | Mean (SD) | Mean (SD) | Mean (SD) | Mean (SD) | Mean (SD) |
| **Installation** | (˚F /˚C) | (˚F /˚C) | (˚F /˚C) | (˚F /˚C) | (˚F /˚C) | (˚F /˚C) |
| Fort Bliss, TX | 65.67 (0.87) | 63.3 (0.83) | 56.23 (0.78) | 80.48 (1.21) | 78.46 (1.04) | 67.96 (0.81) |
| | 18.7 (0.48) | 17.39 (0.46) | 13.46 (0.43) | 26.93 (0.67) | 25.81 (0.58) | 19.98 (0.45) |
| Fort Benning, GA | 66.3 (1.08) | 67.02 (1.18) | 63.71 (1) | 79.64 (1.39) | 82.53 (1.69) | 76.43 (0.96) |
| | 19.05 (0.6) | 19.45 (0.66) | 17.61 (0.55) | 26.47 (0.77) | 28.07 (0.94) | 24.68 (0.53) |
| Fort Bragg, NC | 62.52 (1.19) | 62.85 (1.24) | 60.19 (1) | 77.14 (1.71) | 79.27 (1.91) | 74.26 (1.2) |
| | 16.95 (0.66) | 17.14 (0.69) | 15.66 (0.56) | 25.08 (0.95) | 26.26 (1.06) | 23.48 (0.67) |
| Fort Campbell, KY | 59.02 (1.31) | 59.36 (1.39) | 57.15 (1.13) | 74.82 (1.6) | 76.84 (1.94) | 72.64 (1.26) |
| | 15.01 (0.73) | 15.2 (0.77) | 13.97 (0.63) | 23.79 (0.89) | 24.91 (1.08) | 22.58 (0.7) |
| Fort Hood, TX | 69.41 (1.27) | 69.66 (1.15) | 64.55 (0.84) | 83.7 (1.91) | 85.58 (1.57) | 76.84 (0.65) |
| | 20.79 (0.71) | 20.92 (0.64) | 18.08 (0.47) | 28.72 (1.06) | 29.77 (0.87) | 24.91 (0.36) |
| Fort Jackson, SC | 63.59 (1.12) | 64.1 (1.2) | 61.56 (1) | 77.96 (1.63) | 80.35 (1.77) | 75.3 (1.05) |
| | 17.55 (0.62) | 17.83 (0.66) | 16.42 (0.56) | 25.53 (0.91) | 26.86 (0.99) | 24.05 (0.59) |
| Fort Leonard Wood, MO | 56.51 (1.51) | 56.58 (1.49) | 54.53 (1.19) | 73.28 (1.88) | 74.69 (2.02) | 70.81 (1.34) |
| | 13.62 (0.84) | 13.66 (0.83) | 12.52 (0.66) | 22.93 (1.05) | 23.72 (1.12) | 21.56 (0.74) |
| Fort Polk, LA | 67.11 (1) | 68.25 (1.04) | 64.91 (0.89) | 79.81 (1.65) | 83.2 (1.66) | 77.1 (0.84) |
| | 19.51 (0.56) | 20.14 (0.58) | 18.28 (0.5) | 26.56 (0.92) | 28.45 (0.92) | 25.05 (0.47) |
| Fort Riley, KS | 55.93 (1.82) | 55.59 (1.7) | 52.68 (1.28) | 74.85 (2.27) | 75.5 (2.11) | 70.19 (1.22) |
| | 13.29 (1.01) | 13.11 (0.94) | 11.49 (0.71) | 23.8 (1.26) | 24.16 (1.17) | 21.22 (0.68) |
| Fort Stewart, GA | 67.82 (0.95) | 69.04 (1.11) | 65.84 (1.01) | 79.7 (1.01) | 83.25 (1.36) | 77.29 (0.83) |
| | 19.9 (0.53) | 20.58 (0.62) | 18.8 (0.56) | 26.5 (0.56) | 28.47 (0.75) | 25.16 (0.46) |

Shaded cells indicate a positive linear regression slope at α = 0.05, i.e. a warming trend.

Other studies have characterized annual-scale risks with different approaches and outcomes. A multi-decadal morbidity study in India found a substantial increase in the probability of greater than 100 heat-related deaths occurring due to a 0.5˚C (0.9˚F) increase in mean summer temperature (from 13% to 32%) [31]. A study of heat stress risks for the pilgrimage to Mecca, Saudi Arabia modelled 1.5˚C (2.7˚F) and 2˚C (3.6˚F) mean temperature increases and projected heat stroke risk ratios in the 5.0 to 10.0 range, respectively, during high wet-bulb temperature periods [32].

In this study, we identified positive decadal trends among indices of heat and humidity and among HSI outcomes at active-duty CONUS Army installations. We found overall positive association patterns for ambulatory and reportable event outcomes with temperature, HI, and WBGT indices in absolute and relative annual measures. The largely null finding for hospitalization associations may be due to the low number of annual HSI admissions, as five of the ten CONUS installations averaged fewer than five HSI hospitalizations per year (Table 2). Considering the relative rarity of diagnosed HSI hospitalizations, the availability of ambulatory encounter and reportable event data adds substantial value for the characterization of HSI morbidity that is difficult to match with data sources outside the military health system.

Among the combinations of indices evaluated in our models, HI- and WBGT-based indices generally returned higher RRs than temperature-based indices for each degree increase, possibly due to the incorporation of relative humidity in these measures. We hypothesized that models of indices aggregated over the heat season would return stronger RRs than those aggregated over the full calendar year; however, the opposite effect was observed for most pairings.

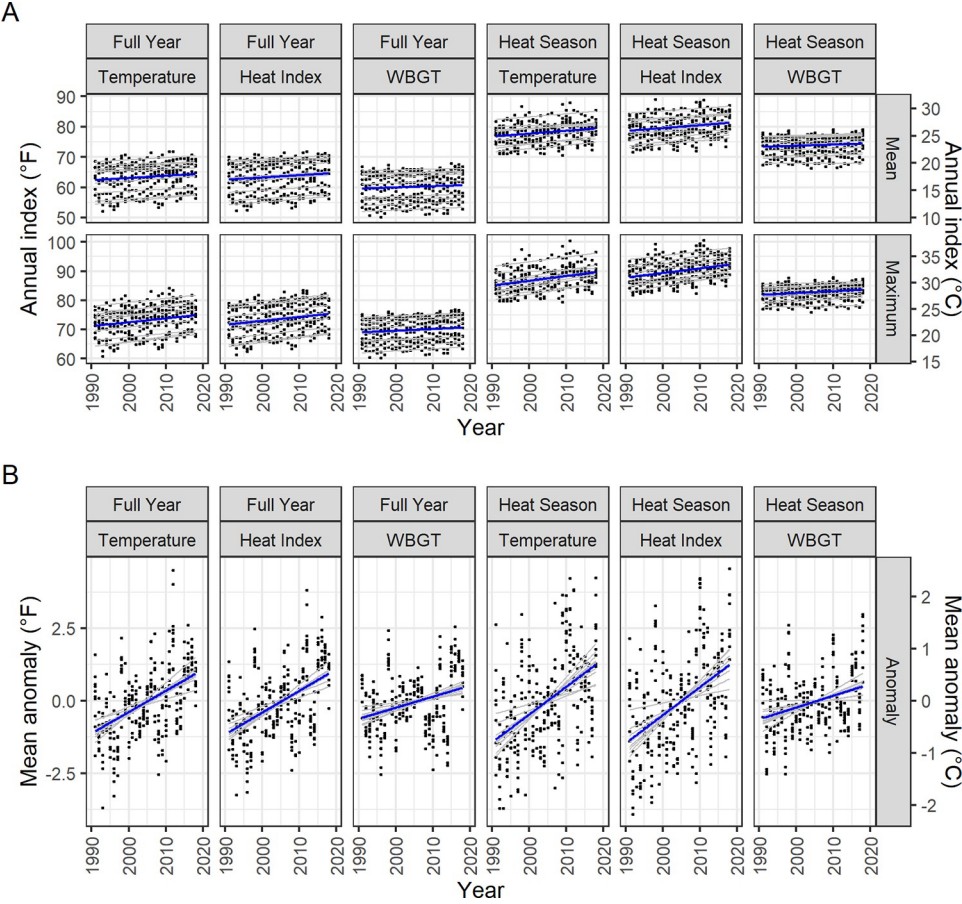

**Fig 3. Annual trends of full-year and heat season degree-based indices of heat at 10 CONUS U.S. Army installations, 1991–2018.** Panel A displays annual means of mean and maximum daily index values. Panel B displays annual means of daily anomalies, relative to location-based 1990–2019 climatologies. Thick blue lines depict the linear trend through all data points. Thin gray lines depict location-specific trends. Indices are in ˚F on the left y-axis and in ˚C on the right y-axis.

This finding furthers evidence for expanding the boundaries of the traditional heat season and incorporating prevention efforts throughout the year [33, 34]. Among exposure measure types, indices based on counts of hours over selected thresholds returned weak or null RRs (set to one-hour increments). Model results for counts of days above one standard deviation relative to the daily climate normal (set to one-day increments) returned positive RRs for ambulatory outcomes and mixed null and positive RRs for hospitalizations and reportable events. The magnitude of the findings does not necessarily indicate that groups of indices are more correct or appropriate than others; rather, they may be more sensitive to detecting associations at an annual scale in support of our hypotheses that heat indices and HSI outcomes are positively related. The unit scales and increments also vary among indices, altering magnitude, but not directionality. There are no suspected mechanisms for negative associations; associations in either direction may be due to chance or impacted by other unmeasured factors.

In these analyses, we assumed that the frequencies and intensities of outdoor training events remained consistent over time for each location and that population-level risk factors did not fluctuate. We made these assumptions considering that the major unit compositions and training and operational mission sets at the selected sites remained mostly stable over the evaluated timeframe. Challenges to this assumption occur from installation population

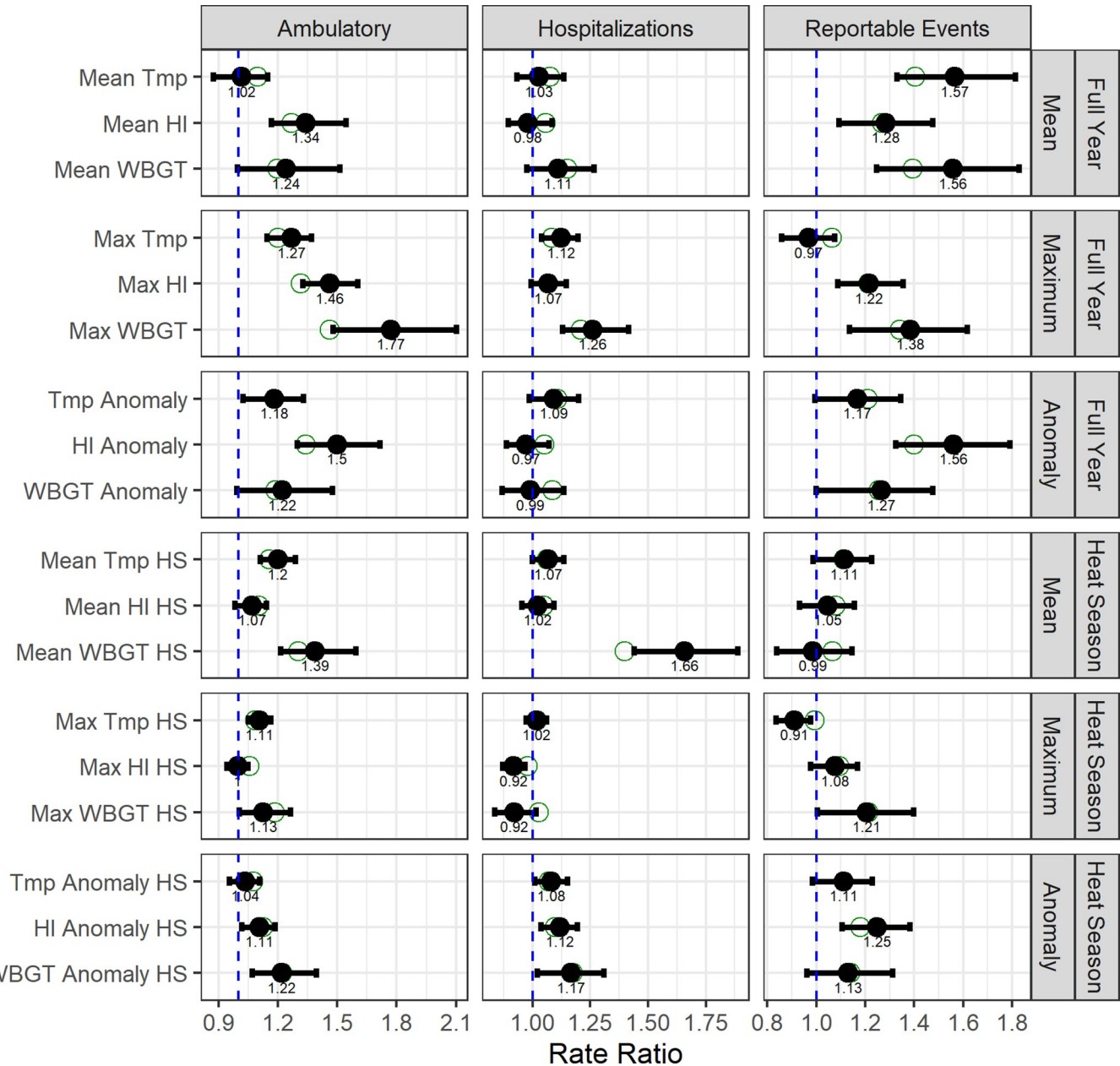

**Fig 4. Rate ratios for full-year and heat season indices of heat and HSI encounters at 10 CONUS U.S. Army installations.** RRs per 1 degree increase in annual index of heat (mean of daily means) from 2-year block bootstrap negative binomial models with basic (empirical) confidence intervals based on 2,000 replicates, controlling for installation-level effects. Solid points reflect the mean of bootstrap estimates and unfilled points reflect the original sample (non-bootstrap) estimate. The dashed blue vertical lines depict a RR of 1.0, indicative of no increased risk.

changes due to extended large unit overseas deployments and organizational changes, such as the movement of the Armor School from Fort Knox, KY to Fort Benning, GA in 2011. Other notable changes included an extension of basic combat training length from eight to nine weeks in 2000 and an increase in active-duty population end strength between 2002 (approximately 485,000)– 2011 (over 560,000) [35]. Shifts in military demographics over the study period reflected increased proportions of female servicemembers and decreased proportions of non-Hispanic white servicemembers relative to other racial and ethnic groups [36]. Trends

**Table 4. Annual scale index-HSI outcome rate ratio 95% confidence interval from 2-year block bootstrap negative binomial models.**

| Index Name | Averaging Period | Index Scale | Ambulatory | Hospitalizations | Reportable Events |
|---|---|---|---|---|---|
| **Mean** | | | | | |
| *Mean Tmp* | Full Year | Temperature | 1.096 (0.872, 1.148) | 1.076 (0.932, 1.133) | 1.404 (1.328, 1.813) |
| *Mean HI* | Full Year | Heat Index | 1.269 (1.168, 1.544) | 1.057 (0.893, 1.084) | 1.267 (1.091, 1.475) |
| *Mean WBGT* | Full Year | WBGT | 1.196 (0.994, 1.511) | 1.149 (0.973, 1.265) | 1.395 (1.246, 1.827) |
| *Mean Tmp HS* | Heat Season | Temperature | 1.157 (1.11, 1.287) | 1.063 (0.998, 1.133) | 1.113 (0.987, 1.227) |
| *Mean HI HS* | Heat Season | Heat Index | 1.105 (0.983, 1.142) | 1.049 (0.951, 1.091) | 1.08 (0.931, 1.156) |
| *Mean WBGT HS* | Heat Season | WBGT | 1.303 (1.214, 1.593) | 1.399 (1.439, 1.886) | 1.065 (0.838, 1.145) |
| **Maximum** | | | | | |
| *Max Tmp* | Full Year | Temperature | 1.2 (1.144, 1.368) | 1.083 (1.038, 1.195) | 1.065 (0.857, 1.074) |
| *Max HI* | Full Year | Heat Index | 1.316 (1.324, 1.601) | 1.069 (0.993, 1.144) | 1.212 (1.086, 1.354) |
| *Max WBGT* | Full Year | WBGT | 1.462 (1.479, 2.099) | 1.21 (1.128, 1.413) | 1.342 (1.134, 1.616) |
| *Max Tmp HS* | Heat Season | Temperature | 1.085 (1.05, 1.163) | 1.018 (0.971, 1.06) | 0.992 (0.835, 0.977) |
| *Max HI HS* | Heat Season | Heat Index | 1.058 (0.942, 1.049) | 0.98 (0.869, 0.965) | 1.095 (0.976, 1.168) |
| *Max WBGT HS* | Heat Season | WBGT | 1.185 (1.005, 1.261) | 1.029 (0.834, 1.013) | 1.212 (1.004, 1.396) |
| **Anomaly** | | | | | |
| *Tmp Anomaly* | Full Year | Temperature | 1.182 (1.023, 1.328) | 1.108 (0.985, 1.199) | 1.211 (0.994, 1.344) |
| *HI Anomaly* | Full Year | Heat Index | 1.341 (1.297, 1.714) | 1.053 (0.887, 1.07) | 1.399 (1.324, 1.79) |
| *WBGT Anomaly* | Full Year | WBGT | 1.188 (0.992, 1.475) | 1.085 (0.868, 1.133) | 1.257 (0.999, 1.475) |
| *Tmp Anomaly HS* | Heat Season | Temperature | 1.075 (0.955, 1.107) | 1.07 (1.008, 1.149) | 1.113 (0.982, 1.227) |
| *HI Anomaly HS* | Heat Season | Heat Index | 1.125 (1.018, 1.184) | 1.097 (1.035, 1.193) | 1.18 (1.104, 1.381) |
| *WBGT Anomaly HS* | Heat Season | WBGT | 1.223 (1.071, 1.393) | 1.175 (1.018, 1.309) | 1.139 (0.962, 1.311) |
| **Hours Count** | | | | | |
| *Hrs Tmp > 90* | Full Year | Temperature | 1 (0.999, 1) | 1 (0.999, 1) | 1 (0.999, 1.001) |
| *Hrs Tmp > 100* | Full Year | Temperature | 1 (0.999, 1.002) | 1 (0.998, 1) | 1.003 (1.004, 1.007) |
| *Hrs HI > 90* | Full Year | Heat Index | 1.001 (1.001, 1.002) | 1 (1, 1.001) | 1.001 (0.999, 1.001) |
| *Hrs HI > 105* | Full Year | Heat Index | 1.004 (0.999, 1.008) | 0.999 (0.991, 1) | 1 (0.991, 1.002) |
| *Hrs WBGT > 85* | Full Year | WBGT | 1.002 (1.001, 1.004) | 1.001 (0.999, 1.001) | 1.002 (1, 1.004) |
| *Hrs WBGT > 90* | Full Year | WBGT | 1.003 (0.995, 1.004) | 1.002 (0.996, 1.005) | 0.999 (0.987, 0.999) |
| **Days Count** | | | | | |
| *Days Tmp > 1 SD* | Full Year | Temperature | 1.013 (1.004, 1.023) | 1.002 (0.982, 1.003) | 1.033 (1.034, 1.063) |
| *Days HI > 1 SD* | Full Year | Heat Index | 1.016 (1.008, 1.03) | 1.01 (0.999, 1.02) | 1.006 (0.986, 1.01) |
| *Days WBGT > 1 SD* | Full Year | WBGT | 1.018 (1.009, 1.044) | 1.008 (0.992, 1.017) | 1.023 (1.011, 1.046) |
| *Days Tmp > 1 SD HS* | Heat Season | Temperature | 1.02 (1.005, 1.046) | 1.006 (0.985, 1.02) | 1.008 (0.975, 1.024) |
| *Days HI > 1 SD HS* | Heat Season | Heat Index | 1.026 (1.01, 1.048) | 1.036 (1.028, 1.08) | 1.019 (1.001, 1.053) |
| *Days WBGT > 1 SD HS* | Heat Season | WBGT | 1.079 (1.079, 1.2) | 1.02 (0.948, 1.065) | 1.009 (0.98, 1.084) |

in overall fitness levels and body composition represented growing HSI risk factors [37–39]. We additionally assumed that HSI prevention measures, including annual safety training requirements and monitoring of WBGT heat categories with associated work-rest cycle and hydration recommendations, had not meaningfully varied over the study time-course [3]. The block bootstrap method to adjust CIs for time trends, along with the inclusion of installation indicator variables, mitigate these potential changes within and between installations over time.

It is necessary to consider whether other time-varying trends account for changes in reported HSI rates. Changes in access to care, case definitions, and reporting systems and procedures can all contribute to long-term trends in the outcomes we studied. We observed impacts from such changes when comparing the rates of all ICD-coded illnesses and injuries

over time, especially for ambulatory rates. The block bootstrap method adjusts for such serial correlation in outcomes. Another limitation with our annually aggregated health outcome counts is that we were unable to discern incident cases from follow-up encounters. The ambulatory counts and rates are therefore elevated above incidence-based case definition levels. However, in this aspect, these data serve to provide representation of the overall burden on the healthcare system from HSIs.

This study assesses the long-term impacts of environmental changes on direct heat-related morbidity; however, it lacks the within-year temporal resolution needed to inform day-to-day or operational level decisions. Important short-term exposure parameters include the intensity, duration, and timing in season of extreme heat events [40]. Further study of HSI morbidity among physically active populations with outdoor environmental exposures may expand upon the short-term exposure-response relationship between heat and humidity indices and daily outcomes, considering lagged and non-linear effects and controlling for individual-level risk factors [13].

## Conclusion

U.S. Army CONUS installations have broadly experienced rising temperature conditions and increased rates of HSI morbidity over the past two to three decades, despite having long-standing, institutionalized heat stress control guidelines and regulations. In this study, we determine that temperature, HI, and WBGT indices are positively associated with rates of ambulatory encounters and reportable events, controlling for installation-levels effects and accounting for potential confounding by long-term trends in the outcomes and exposures. The annual-scaled rate ratios and their uncertainties can be applied to climate projections for a wide range of thermal indices to estimate future HSI burden and impacts to medical readiness. In an example application, the ambulatory HSI RR is 1.16 for a 1°F (0.55°C) increase in mean temperature between May and September. In 2018, the active-duty population of approximately 204,291 at the included ten CONUS installations reported 3,612 ambulatory HSI encounters. Applying this effect estimate, a 1°F (0.55°C) increase in the heat season mean annual temperature would lead to a projected increase to 4,190 HSI ambulatory encounters (+578 cases) in the absence of additional adaptations or control measures. Application of the findings can extend to physically active members of the general population for climate change impact and risk analysis, while acknowledging that some characteristics of exposure and utilization of care are unique to the military. The observed, increasing HSI outcome trend signals the need for renewed emphasis on adaptation measures to counter heat stress risks. Effective prevention strategies should span socio-ecological framework levels with interactions involving individuals, interpersonal relationships, organizations, communities, and society, involving leaders at all echelons as well as the medical community [41]. In military terminology, consideration of each dimension of the DOTMLPF-P (Doctrine, Organization, Training, Materiel, Leadership, Personnel, Facilities, and Policy) analytical framework is also relevant to the identification of targets for preventive action [42]. Advancements in research and technology may enhance identification of heat stress risk factors, optimize physical conditioning and acclimation training, improve recognition of early heat casualty warning signs, and provide more comprehensive monitoring of environmental conditions.

## Acknowledgments

The opinions and assertions expressed herein are those of the authors and do not necessarily reflect the official policy or position of the Uniformed Services University or the Department of Defense.

## Author Contributions

**Conceptualization:** Stephen A. Lewandowski, Marianthi-Anna Kioumourtzoglou, Jeffrey L. Shaman.

**Data curation:** Stephen A. Lewandowski.

**Formal analysis:** Stephen A. Lewandowski.

**Investigation:** Stephen A. Lewandowski.

**Methodology:** Stephen A. Lewandowski, Marianthi-Anna Kioumourtzoglou, Jeffrey L. Shaman.

**Software:** Stephen A. Lewandowski.

**Supervision:** Marianthi-Anna Kioumourtzoglou, Jeffrey L. Shaman.

**Validation:** Stephen A. Lewandowski.

**Visualization:** Stephen A. Lewandowski.

**Writing – original draft:** Stephen A. Lewandowski.

**Writing – review & editing:** Stephen A. Lewandowski, Marianthi-Anna Kioumourtzoglou, Jeffrey L. Shaman.

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
