## [Decision Letter · Decision Letter 0]

1 Apr 2022

PONE-D-22-02473Heat Stress Illness Outcomes and Annual Indices of Outdoor Heat at U.S. Army InstallationsPLOS ONE

Dear Dr. Lewandowski,

Thank you for submitting your manuscript to PLOS ONE. After careful consideration, we feel that it has merit but does not fully meet PLOS ONE’s publication criteria as it currently stands. Therefore, we invite you to submit a revised version of the manuscript that addresses the points raised during the review process.

We look forward to receiving your revised manuscript.

Kind regards,

Yanping Yuan

Academic Editor

PLOS ONE

Journal Requirements:

“S.L was supported by the U.S. Army Long Term Health Education and Training (LTHET) program. J.S. and Columbia University disclose partial ownership of SK Analytics. J.S. also reports receiving consulting fees from Merck and BNI.”

“The opinions and assertions expressed herein are those of the authors and do not necessarily reflect the official policy or position of the Uniformed Services University or the Department of Defense. S.L was supported by the U.S. Army Long Term Health Education and Training (LTHET) program. J.S. and Columbia University disclose partial ownership of SK Analytics. J.S. also reports receiving consulting fees from Merck and BNI. The funders had no role in study design, data collection and analysis, decision to publish, or preparation of the manuscript.”

Reviewers' comments:

Reviewer's Responses to Questions

**Comments to the Author**

1. Is the manuscript technically sound, and do the data support the conclusions?

Reviewer #1: Yes

Reviewer #2: Partly

2. Has the statistical analysis been performed appropriately and rigorously? 

Reviewer #1: Yes

Reviewer #2: Yes

3. Have the authors made all data underlying the findings in their manuscript fully available?

Reviewer #1: Yes

Reviewer #2: Yes

4. Is the manuscript presented in an intelligible fashion and written in standard English?

Reviewer #1: Yes

Reviewer #2: Yes

5. Review Comments to the Author

Reviewer #1: Overall this is an interesting study specially related to the occupational health. There are previous studies on this scope but more are on the industrial workers. This study is based on army which make it more interesting. The industrial workers are protected by the OSH regulations. How about for the Army officers? This is an interesting question that can be answered by author.

Please refer the attachment for the suggestions.

Reviewer #2: General Comments:

Overall, a well-written paper on a topic of interest to biometeorology, climate, and health research. Some methodological questions are noted below. However the primary concerns center on the use of NLDAS air temperature estimates as the weather conditions to compare to health data. Secondly, the time scale of the health data is unclear. Are the incidents being reported as the day of occurrence or is the data only being reported in aggregate as totals for each year (and/or each summer)? The ambulatory rates are the primary focus of the results in the abstract, but it is a bit unclear as to whether or not the ambulatory rates have accounted for the background increase in ambulatory rates. Another point would be how narrow the research application is compared to the potential areas of research that would benefit from this study. The focus is on implications for military personnel, but the research could be applicable to the general population as well as to athletic communities.

Specific comments:

Line 30 — why use fahrenheit? Is that the standard in the US Military? Most science research journals expect units of temperature to be in Celsius (or Kelvin), including PLoS One (https://journals.plos.org/plosone/s/submission-guidelines). Please transfer results to celsius or Kelvin

Line 35 – why should us military heat stress matter to the general international population? What are some of the implications of military heat stress for the general population?

Line 44 – can you broaden out a little and connect this to other subgroups of the population, like athletes who also tend to have increased time outdoors, with high physical exertion levels?

Line 58 – are the aggregate values including max/min values? If the US Army uses WBGT frequently, why not use their data?

Line 60 – annual HSI morbidity outcomes are important, but are higher temporal resolution health data (such as daily reports of HSI morbidity) available?

Line 66 – Are the data being normalized for any changes in the underlying population over the time period? Was there an increase/decrease in military population over the time period?

Line 84-85 – could you rephrase and elaborate a bit. You haven’t identified the ten locations yet. Is Fort Irwin on the list? Or is it the eleventh and thus not included in the study? It’s a bit unclear. A table of the top fifteen with the number of HSI cases per location might be helpful to explain why only the top ten were included.

Line 98 – How is 2-meter air temperature calculated from the remotely sensed NLDAS-2 data? Remotely sensed data cannot directly measure air temperature and LST is a known proxy (but a poor one) for air temperature and the calculation varies from product to product.

Line 104-107 – why was the data from the 14th Weather Squadron not used directly to study the relationship between HSI and WBGT? Are there forts without weather data?

Line 126 – Is this a yearly summation? Or a finer temporal scale?

Line 134 – Why is Fort Bliss selected as the reference station? Why not another location higher/lower on the list?

Line 150 – why was the number of replications in the bootstrapping lower for non-select indices? Is there a method to determine which indices to replicate with 10,000 repetitions?

Line 158-159 – are you accounting for any policy or population changes over the study period?

Table 1 – Would prefer some shading to identify which locations/conditions are statistically significantly positive/negative. This is also the first time we are given a glimpse into the list of locations. Is there any correlation between the location of the fort (i.e., background climatology) and the HSI?

Line 181 – 1,040 index-installation pairs… meaning 1,040 incidents? If so, that would only be 10 incidents per year at any given location. Is that sufficient (even with bootstrapping) to make statistical conclusions?

Table 2 – would also benefit from shading to help in identifying statistical significance (and direction of relationship).

Line 195 – how are the installation-specific effects controlled?

Line 205 – less than half of the pairs have positive relationships. Is this statistically significant?

Line 232 – since hospitalization is so unusual, has a weak statistical power, and thus is largely a null finding, why include this in the study? Maybe adding some language about the basic summary statistics of the HSI incidents would be helpful to clarify.

Line 238-239 – good point of clarity on why various indices of heat perform differently.

Line 244-246 – I think this is an interesting point, but isn’t a direct conclusion from this study. Could you include some citations of other research that has shown this to support the claim?

Line 256-257 – has the population itself changed (even if the demographics haven’t)?

Line 266-268 – given that ambulatory rates have increased regardless of ICD coded illness or injury, was this trend controlled for in your analysis?

Figure 1 – the # of ambulatory incidents was ~5 times larger than reportable events… Shouldn’t every ambulatory incident be reported? Please clarify for your non-military audience.

Line 275-276 – if the data lack within-year temporal resolution, how do you calculate the summer (May-September) versus annual rate of HSI?

Line 277-281 – Could this kind of research help inform an upper -threshold of heat tolerance for the US population? What does this say for populations (military and civilian) not located in these regions of the CONUS (say Hawai’i or Arizona)?

Line 290 – the strongest relationship for ambulatory HSI is also the rate with the greatest general change over time (regardless of health outcome)? This is a bit concerning as it is unclear how the change in ambulatory rates was controlled for over the study period.

Line 292 – this is the first time the total population of the military at these locations for a given year is provided. This should be part of the methods or early descriptive results. It is useful here as well to make a note of what the RR would produce in terms of actual ambulatory calls.

6. PLOS authors have the option to publish the peer review history of their article (what does this mean?). If published, this will include your full peer review and any attached files.

Reviewer #1: No

Reviewer #2: **Yes: **Peter J. Crank

---

## [Author Response · Author response to Decision Letter 0]

7 Jul 2022

Author’s note:

In addition to the revisions made to address the reviewer comments, we re-analyzed the data after discovering misassigned climate data at multiple locations. Following the correction, the model results and conclusions were not affected to a large degree since overall year-to-year variability was still captured by the data points. The changes are reflected in the Table 2 summary of annual heat indices. 

Additionally, this revision reports a more selective set of indices and incorporates the Supporting Materials tables into the main text. 

Reviewer #1: 

1. Overall this is an interesting study specially related to the occupational health. There are previous studies on this scope but more are on the industrial workers. This study is based on army which make it more interesting. The industrial workers are protected by the OSH regulations. How about for the Army officers? This is an interesting question that can be answered by author. 

Under OSH regulation, workers has been protected by this regulation in regards to heat exposure. Is there any related regulation for U.S army? or any existing risk control?

OSHA has moved forward a rulemaking process for a heat-specific workplace standard, but there is not currently a federal regulation in place. In April 2022, OSHA published an Instruction for a National Emphasis Program on heat-related hazards in support of Executive Order (EO) 14008, “Tackling the Climate Crisis at Home and Abroad” (https://www.osha.gov/sites/default/files/enforcement/directives/CPL_03-00-024.pdf). 

The military has had regulations in place since the post-World War II era regarding prevention of heat casualties. The policies include heat stress risk tables based on WBGT. These are described in the Introduction and Discussion sections. 

2. It will good for the future reader if the author could provide some info on the type of activity done by this active-duty soldiers. Maybe their work activity such as how long their spend in a day in this condition. Any existing risk control? or is it confidential to be revealed?

We have added text and citations in the Introduction on the physically demanding tasks encountered in the military (end of first paragraph). Yes, there is existing risk control, which has remained consistent throughout the study period. This control is described in the Introduction and Discussion sections.

3. Do we have any info for this in regard to the duration of this study? 1991 to 2018.

Data for the annual-scale warming trends at the study location are provided in Table 2 and Figure 2.

4. This study is based on secondary data?

Yes. We used cross-sectional, aggregate counts of HSI outcomes, queried and constructed from the Defense Medical Surveillance System, in this study. The NLDAS-2 modelled estimates that we compiled for annual indices were developed by a NASA/NOAA-led consortium. 

5. Is possible author could provide some brief info on this database as it will be some clearer picture for the future reader? Either in this section or in the introduction section.

We have added additional information on access to the database and outcome-type details in the first paragraph under “Materials and Methods: Health Outcome Data”: “Hospitalization and ambulatory data include encounters from Department of Defense (DoD) and non-DoD treatment facilities. Reportable events are defined in the Armed Forces Reportable Medical Events Guidelines and Case Definitions and represent conditions that pose a significant threat to public health and military operations [14]. The DMED application is accessible through the Armed Forces Health Surveillance Division at https://www.health.mil/dmed/ for authorized users and validated medical researchers [15].”

6. Could author provide this process with a systematic review flow diagram?

We have added Fig 1 schematic (clinical classification codes and included outcome types by year).

7. How is the reliability of data from this stations?

These military weather station data are reliable (used for airfield operations and reported to the National Climatic Data Center). However, we removed this reference to 14th Weather Squadron from the revision since it was used for background comparison and not for primary analyses. A concern with station data is the variable distance from the included Army installations. 

8. Again, it will be good for the future reader if the author could also provide this information in a flow diagram (as figure).

We have added Table 1 (classification of included annual indices) to outline the categories of selected annual indices.

9. Do we have any similar previous study?

No, not to our knowledge for a closely matched study design and exposure-outcome pairing. A new first paragraph was added to the Discussion section addressing related studies.

10. Is there any risk control taken during this past 2 or 3 decades?

Yes, we have added some more information to the first sentence in the Conclusion section: “despite having long-standing, institutionalized heat stress control guidelines and regulations.”

11. What are existing risk control and commended risk control for this study?

We have added the following closing statements of the Conclusions: “The observed, increasing HSI outcome trend signals the need for renewed emphasis on adaptation measures to “concur the heat”. Prevention strategies span socio-ecological framework levels with interactions involving individuals, interpersonal relationships, organizations, the community, and society, involving leaders at all echelons as well as the medical community [34]. Consideration of each dimension of the DOTMLPF-P (Doctrine, Organization, Training, Materiel, Leadership, Personnel, Facilities, and Policy) analytical framework is also relevant to the identification of targets for preventive action [35]. Advancements in research and technology may enhance identification of heat stress risk factors, optimize physical conditioning and acclimation training, improve recognition of early heat casualty warning signs, and provide more comprehensive monitoring of environmental conditions.” 

Reviewer #2: General Comments:

1. Overall, a well-written paper on a topic of interest to biometeorology, climate, and health research. Some methodological questions are noted below. However the primary concerns center on the use of NLDAS air temperature estimates as the weather conditions to compare to health data. Secondly, the time scale of the health data is unclear. Are the incidents being reported as the day of occurrence or is the data only being reported in aggregate as totals for each year (and/or each summer)?

The incidents in this paper are all aggregates as totals for each year. 

The ambulatory rates are the primary focus of the results in the abstract, but it is a bit unclear as to whether or not the ambulatory rates have accounted for the background increase in ambulatory rates. Another point would be how narrow the research application is compared to the potential areas of research that would benefit from this study. The focus is on implications for military personnel, but the research could be applicable to the general population as well as to athletic communities.

This research is also relevant beyond the military (translation of the findings, methodologies, and climate trends at locations near Army installations). We have added or updated text on civilian population applications in the Abstract, Introduction, and Conclusion sections.

In the Conclusion, we state: “Applications of the findings can extend to physically active members of the general population for climate change impact and risk analysis, while acknowledging that some characteristics of exposure and utilization of care are unique to the military.”

Specific comments:

2. Line 30 — why use fahrenheit? Is that the standard in the US Military? Most science research journals expect units of temperature to be in Celsius (or Kelvin), including PLoS One (https://journals.plos.org/plosone/s/submission-guidelines). Please transfer results to celsius or Kelvin

Fahrenheit is the common standard for use in the U.S. Army, and is the scale used on WBGT Heat Category tables (along with National Weather Service heat index tables). 

We now additionally provide Celsius units throughout the text, tables, and figure axes scales.

3. Line 35 – why should us military heat stress matter to the general international population? What are some of the implications of military heat stress for the general population?

We have added some text on generalizability in the Introduction paragraph: “Military heat stress exposures may be broadly generalizable to civilian populations with a similar age distribution, exposure to outdoor conditions, and exertional levels, including athletic and occupational settings [8]. However, specific prevention guidelines should account for risk factor differences between groups [9].” 

4. Line 44 – can you broaden out a little and connect this to other subgroups of the population, like athletes who also tend to have increased time outdoors, with high physical exertion levels?

We have added text broadening the applications: “Military heat stress exposures may be broadly generalizable to civilian populations with a similar age distribution, exposure to outdoor conditions, and exertional levels, including athletic and occupational settings [8]. However, specific prevention guidelines should account for risk factor differences between groups [9].” 

5. Line 58 – are the aggregate values including max/min values? If the US Army uses WBGT frequently, why not use their data?

In this study, we compiled mean (annual mean of mean daily) and maximum (annual mean of maximum daily) values, but not minimum. U.S. Army personnel apply WBGT values in real-time at the small-unit level; however, the Army does not systematically collect or record WBGT data across locations. 

6. Line 60 – annual HSI morbidity outcomes are important, but are higher temporal resolution health data (such as daily reports of HSI morbidity) available?

When this study was initiated, daily outcome data were not available to the authors, and the study was designed to match cross-sectional annual counts with annual indices. Records of de-identified medical encounters at a daily scale were later obtained through a different system and were assessed in a separate paper with a different approach: Lewandowski SA, Shaman JL. Heat stress morbidity among US military personnel: Daily exposure and lagged response (1998-2019). Int J Biometeorol. 2022;66: 1199–1208. doi:10.1007/s00484-022-02269-3. The findings from these two studies serve different purposes (long-term versus short-term responses to heat).

7. Line 66 – Are the data being normalized for any changes in the underlying population over the time period? Was there an increase/decrease in military population over the time period?

The populations for each location and for each year are included in the offset term for the model to normalize for changes (described in Statistical Analysis). 

We have added in Materials and Methods/ Health Outcome Data: “The rates are based on active component servicemember populations at each location for each year.”

We have also added in the Results (end of first paragraph): “The overall active component population from the 10 included installations varied over time. The total population increased between 1991–2011 and decreased between 2011–2018, ranging from 176,490 in 1991 to 249,915 in 2011.” 

8. Line 84-85 – could you rephrase and elaborate a bit. You haven’t identified the ten locations yet. Is Fort Irwin on the list? Or is it the eleventh and thus not included in the study? It’s a bit unclear. A table of the top fifteen with the number of HSI cases per location might be helpful to explain why only the top ten were included.

We have rephrased to clarify – Fort Irwin was excluded. Table 2 provides a list of included installations and outcome statistics.

9. Line 98 – How is 2-meter air temperature calculated from the remotely sensed NLDAS-2 data? Remotely sensed data cannot directly measure air temperature and LST is a known proxy (but a poor one) for air temperature and the calculation varies from product to product.

2-meter air temperature is directly provided as one of the 11 NLDAS-2 land-surface forcing fields (no additional user calculations were performed). NLDAS-2 simulations combine satellite data with reanalysis model data derived from multi-source observations, including station data. The NLDAS-2 model applies a vertical adjustment using a standard lapse rate for air temperature. 

10. Line 104-107 – why was the data from the 14th Weather Squadron not used directly to study the relationship between HSI and WBGT? Are there forts without weather data?

The Air Force data, as provided with WBGT estimates, did not cover the full study period (available from 2008–2018). Although data were available for the large US Army forts, another concern was the distance of some stations away from the center of the installations (over 50 km for Fort Bliss). We have removed the mention of 14th Weather Squadron data from the revised manuscript to avoid confusion. 

11. Line 126 – Is this a yearly summation? Or a finer temporal scale?

The indices were all aggregated at a yearly level. We have added Table 1 to provide further clarification on the compiled indices.

12. Line 134 – Why is Fort Bliss selected as the reference station? Why not another location higher/lower on the list?

We have added: “…due to having the lowest HSI encounter counts among the included sites.” The selection of the categorical reference value does not impact the performance of the model or the results. 

13. Line 150 – why was the number of replications in the bootstrapping lower for non-select indices? Is there a method to determine which indices to replicate with 10,000 repetitions?

We changed and standardized this selection in the revised manuscript. All bootstrap models were run with 2,000 replications and the manuscript has been updated to reflect this change. In the initial run, a core set of indices was picked to examine the effect of a greater number of repetitions on confidence interval size. 

14. Line 158-159 – are you accounting for any policy or population changes over the study period?

The HSI rate trends do account for population change over time (included in the denominators). The rate trends do not account for any policy changes over time. The design of our regression models describing the association between heat indices and outcomes, however, controls for such changes over time along with differences between installations.

15. Table 1 – Would prefer some shading to identify which locations/conditions are statistically significantly positive/negative. This is also the first time we are given a glimpse into the list of locations. Is there any correlation between the location of the fort (i.e., background climatology) and the HSI?

We have changed the footnote notation to shaded cells for positive and negative slopes. The table number has now advanced to Table 2. 

The correlations between background climatologies and HSI rates vary in direction and magnitude amongst the combinations of index types and outcome types, although most are positive. These associations are tested in our models, where we account for confounding by location through the inclusion of indicator variables. 

16. Line 181 – 1,040 index-installation pairs… meaning 1,040 incidents? If so, that would only be 10 incidents per year at any given location. Is that sufficient (even with bootstrapping) to make statistical conclusions?

We apologize for the confusion. This reflected the prior index combinations (104), i.e. measures of heat exposure, multiplied by the number of installations (10), rather than the total number of heat stress incidents. The total number of indices was streamlined in the revision, and this text has been reworded. Figure 3 was also added to display the index time trends. 

17. Table 2 – would also benefit from shading to help in identifying statistical significance (and direction of relationship).

We have added shading to replace the footnote notation; Table 2 is now Table 3; values have been updated following the re-analysis, and Celsius units were added.

18. Line 195 – how are the installation-specific effects controlled?

Installation-specific effects were controlled in the regression models by the inclusion of installation-specific indicator (dummy) variables (described in Statistical Analysis). 

19. Line 205 – less than half of the pairs have positive relationships. Is this statistically significant?

The number or proportion of positive relationships was not intended to be statistically tested due to their different scales and aggregations. These values have changed with the re-analyses, and the results shown in Fig 4 and Table 4 display an overall positive pattern for Ambulatory and Reportable Event outcomes. 

20. Line 232 – since hospitalization is so unusual, has a weak statistical power, and thus is largely a null finding, why include this in the study? Maybe adding some language about the basic summary statistics of the HSI incidents would be helpful to clarify.

Hospitalizations were included because they represent an important outcome in terms of impact on readiness and resource management. The strength of association was not known beforehand, and there remains value in reporting the trends as well as the limitations. 

We have added the following statement to first paragraph in the Results: “Ambulatory events were most reported, with a mean total of 2,081 per year over the assessed period for the included sites. Reportable events averaged 394 per year and hospitalizations averaged 109 per year.”

21. Line 238-239 – good point of clarity on why various indices of heat perform differently.

Thank you! We have retained this statement in the revision.

22. Line 244-246 – I think this is an interesting point, but isn’t a direct conclusion from this study. Could you include some citations of other research that has shown this to support the claim?

We thank the Reviewer for this suggestion. We have added the following citations:

Stacey et al. 2015 – “nearly a third of all exertional heat illness (EHI) was sustained by UK military personnel during non-summer months”

DeGroot, Martin 2015 – “During the investigation period there were 7,827 EHIs, 79% of which occurred during the heat season. However, between locations there was considerable variability in within heat season EHI frequency”

23. Line 256-257 – has the population itself changed (even if the demographics haven’t)?

Yes, fitness levels and obesity rates have changed (noted in the following lines). We have a manuscript in preparation examining demographic and body composition trends related to heat illness. Surprisingly, the average body mass index (BMI) among heat causalities has remained steady over time despite population-level increases.

24. Line 266-268 – given that ambulatory rates have increased regardless of ICD coded illness or injury, was this trend controlled for in your analysis?

The total cases were not directly controlled for in our models per se. However, the block bootstrap resampling approach was selected to mitigate the overall long-term time trends. It would be difficult to disentangle the impact of climate on HSI, even among all-cause morbidity. 

25. Figure 1 – the # of ambulatory incidents was ~5 times larger than reportable events… Shouldn’t every ambulatory incident be reported? Please clarify for your non-military audience.

We have added a description and citation for Reportable Events in the Materials and Methods/Health Outcome Data: “Reportable events are defined in Armed Forces Reportable Medical Events Guidelines and Case Definitions, representing conditions that pose a significant threat to public health and military operations [14]” The reportable event guidelines have more specific case definitions for confirmed heat exhaustion and probable or confirmed heat stroke.

26. Line 275-276 – if the data lack within-year temporal resolution, how do you calculate the summer (May-September) versus annual rate of HSI?

The cases were not matched within year in this study. The different types of annual indices represent: “was the heat season (summer) hot/humid this year” or “was the entire calendar year hot/humid”. Because we do not know the distribution of cases throughout the year, we wanted to test both sets of averaging periods. If larger numbers of cases occurred outside of the heat season months, we would expect more non-differential misclassification bias for the May-September indices. 

27. Line 277-281 – Could this kind of research help inform an upper -threshold of heat tolerance for the US population? What does this say for populations (military and civilian) not located in these regions of the CONUS (say Hawai’i or Arizona)?

Yes, this kind of research has potential to help inform heat tolerance thresholds for the US population; however, this particular study design is better suited to estimate impacts from moderate increases sustained over long timeframes. It provides data points that may be applied to quantified climate impact assessments and resource allocation planning. Daily (or sub-daily)-scale exposure-response models provide more value for determining upper-threshold heat tolerance from observational data.

The broad estimates can be applied to additional CONUS and OCONUS regions, such as Hawai’i or Arizona, with consideration of the generalization limitations that have been described in the Introduction and Discussion. However, it may not be appropriate to apply the findings to extreme hot or cold climate regions. 

28. Line 290 – the strongest relationship for ambulatory HSI is also the rate with the greatest general change over time (regardless of health outcome)? This is a bit concerning as it is unclear how the change in ambulatory rates was controlled for over the study period.

In the revised analysis, we observed comparably strong relationships across index types for Reportable Event HSIs, which is reassuring since reportable event rates were more steady over time than ambulatory. The block bootstrap statistical method controls for non-cause-specific long term time trends. 

29. Line 292 – this is the first time the total population of the military at these locations for a given year is provided. This should be part of the methods or early descriptive results. It is useful here as well to make a note of what the RR would produce in terms of actual ambulatory calls.

The total population is now described in the Results (end of first paragraph): “The overall active component population from the 10 included installations varied over time. The total population increased between 1991–2011 and decreased between 2011–2018, ranging from 176,490 in 1991 to 249,915 in 2011.” 

The example was updated to “Applying this effect estimate, a 1 °F (0.55 °C) increase in the heat season mean annual temperature would lead to a projected increase to 4,190 HSI ambulatory encounters (+578 cases) in the absence of additional adaptations or control measures.”

---

## [Decision Letter · Decision Letter 1]

30 Aug 2022

PONE-D-22-02473R1Heat stress illness outcomes and annual indices of outdoor heat at U.S. Army installationsPLOS ONE

Dear Dr. Lewandowski,

Thank you for submitting your manuscript to PLOS ONE. After careful consideration, we feel that it has merit but does not fully meet PLOS ONE’s publication criteria as it currently stands. Therefore, we invite you to submit a revised version of the manuscript that addresses the points raised during the review process.

We look forward to receiving your revised manuscript.

Kind regards,

Yanping Yuan

Academic Editor

PLOS ONE

Journal Requirements:

Reviewers' comments:

Reviewer's Responses to Questions

**Comments to the Author**

1. If the authors have adequately addressed your comments raised in a previous round of review and you feel that this manuscript is now acceptable for publication, you may indicate that here to bypass the “Comments to the Author” section, enter your conflict of interest statement in the “Confidential to Editor” section, and submit your "Accept" recommendation.

Reviewer #1: All comments have been addressed

Reviewer #2: (No Response)

2. Is the manuscript technically sound, and do the data support the conclusions?

Reviewer #1: Yes

Reviewer #2: Partly

3. Has the statistical analysis been performed appropriately and rigorously? 

Reviewer #1: Yes

Reviewer #2: Yes

4. Have the authors made all data underlying the findings in their manuscript fully available?

Reviewer #1: Yes

Reviewer #2: Yes

5. Is the manuscript presented in an intelligible fashion and written in standard English?

Reviewer #1: Yes

Reviewer #2: Yes

6. Review Comments to the Author

Reviewer #1: Overall this is an interesting study specially related to the occupational health. There are previous studies on this scope but more are on the industrial workers. However, this study is based on army which make it more interesting.

Overall, author have provided the required revision on the raised questions.

I have no other comments.

Reviewer #2: General Comments:

Many points identified by the reviewers have been addressed. Thank you for taking the time to clarify these points. There are a couple of remaining points that I seek clarity on. There seems to be a lot of restrictions in terms of getting good reliable data. My primary concern is the relationship between remotely sensed air temperature data and health impacts. The NLDAS-2 dataset is a bit murky to me and the annual rates of heat-related incidence make it hard to justify how we can assume that these events are occurring more often on hot days. These limitations must be highlighted in the discussion/conclusion. This appears to identify a gap in knowledge due to insufficient data. The lack of station specific data is surprising given the resources the Army has at their disposal to train their soldiers in a variety of circumstances. This should be included as an area of future study/exploration.

Specific comments:

Line 60 – annual HSI morbidity outcomes are important, but are higher temporal resolution health data (such as daily reports of HSI morbidity) available?

A: When this study was initiated, daily outcome data were not available to the authors,

and the study was designed to match cross-sectional annual counts with annual

indices. Records of de-identified medical encounters at a daily scale were later

obtained through a different system and were assessed in a separate paper with a

different approach: Lewandowski SA, Shaman JL. Heat stress morbidity among US

military personnel: Daily exposure and lagged response (1998-2019). Int J

Biometeorol. 2022;66: 1199–1208. doi:10.1007/s00484-022-02269-3. The findings

from these two studies serve different purposes (long-term versus short-term

responses to heat).

Helpful to know. Could a sentence be added to refer to this paper so that readers who might have the same question could be directed to that study?

Line 98 – How is 2-meter air temperature calculated from the remotely sensed NLDAS-2 data? Remotely sensed data cannot directly measure air temperature and LST is a known proxy (but a poor one) for air temperature and the calculation varies from product to product.

A: No additional user calculations were made… with a standard lapse rate estimate for air temperature.

So does the NLDAS-2 do that via going down from another level? Or by making some assumptions about the surface temp to the immediate air above the surface?

This is based on Luo et al., 2003?

This one is still a bit murky and concerning to me.

Line 158-159 – are you accounting for any policy or population changes over the study period?

A: No policy changes were accounted for in the model as the regression model wasn’t set up for those qualitative data.

Could there be a brief mention of whether or not policy changes occurred in the time (even if out of the scope of the study and purview of the model)?

Figure 4 - it is intriguing that the heat season has a lower rate ratio with reportable events.

Given that the health incidents are only noted at an annual rate, it seems to be difficult claim to make that there is *any* inter-annual variability in the data due to heat. Even if logic says that it should be on days with higher heat.

Line 256-257 – has the population itself changed (even if the demographics haven’t)?

Interesting that BMI has remained steady! Language in the discussion doesn’t appear to mention anything about this.

Line 266-268 – given that ambulatory rates have increased regardless of ICD coded illness or injury, was this trend controlled for in your analysis?

A: difficulty in untangling the impact of climate on all ambulatory rates.

Definitely true on the difficulty of untangling, but could be done by having a trend term in the regression for the independent variable(s) as well as the dependent variable. In your regression model, is there a trend term that would control for the increase over time?

Line 275-276 – if the data lack within-year temporal resolution, how do you calculate the summer (May-September) versus annual rate of HSI?

A: cases were not matched within the study. They would expect there to be a non-differential misclassification bias in their heat season indices.

Ok, so this was completed in a more qualitative manner, i.e., this year (or summer) was hot/humid and the other year was not. But this should be clarified to the reader. These “heat season” metrics are presumed to be valid, but cannot be confirmed through this study design.

7. PLOS authors have the option to publish the peer review history of their article (what does this mean?). If published, this will include your full peer review and any attached files.

Reviewer #1: No

Reviewer #2: No

---

## [Author Response · Author response to Decision Letter 1]

14 Oct 2022

General comments:

We selected NLDAS-2 gridded data over station data early in our design due to its spatial coverage, completeness, parameter availability, hourly values, and validation. The application of gridded climate models is growing in public health research. Spangler, Liang, and Wellenius (2022) write: 

The (epidemiologic research) field is moving toward more expansive analyses that use spatially resolved gridded meteorological datasets and alternative assessments of heat, such as wet-bulb globe temperature (https://doi.org/10.1038/s41597-022-01405-3)

We provide further description of NLDAS-2 data in a response below (reference Line 98) and in an added statement in the manuscript.

There are weather stations in proximity to the installations, but the distance from the base centroid varies by site (often co-located with airfields). The addition of station monitoring sites would represent an important area for future study and exploration.

Specific comments:

Line 60 - Yes, this reference is beneficial to highlight, and has now been published. We added to the second Introduction paragraph:

The relationship between daily-scale indices and HSI encounters at military sites was assessed in a separate study [13].

Line 98 - NLDAS-2 fields are calculated from a land surface model which integrates atmospheric observations from meteorological stations, radiosondes, and satellites with land surface states, such as soil moisture, soil temperature and snow cover (Luo, 2003, [2]).

Station data is a major input for 2-meter air temperature. The calculated values are not solely based on remotely sensed data.

We added the following to the paper under “Meteorology Data”:

Its (NLDAS-2) land surface model integrates atmospheric observations from sources including meteorological stations, radiosondes, and satellites to derive land surface states [20].

Line 158-159 – We expanded the Discussion section (4th paragraph) with examples of policy and population changes.

Other notable changes included an extension of basic combat training length from eight to nine weeks in 2000 and an increase in active-duty population end strength between 2002 (approximately 485,000) – 2011 (over 560,000) [35].

Figure 4 - This is an unexpected result. However, there is a considerable number of cases that occur outside of the traditional May-September heat season, which could help explain the finding.

Line 256-257 – Yes, the total active-duty population has fluctuated over time. We added a description of a change following 9/11 in the Discussion, as noted in the prior response. We accounted for these changes by using rates set by each location and year.

increase in active-duty population end strength between 2002 (approximately 485,000) – 2011 (over 560,000) [35].

We updated a demographics statement from “Demographics of age, sex, and ethnicity…did not markedly change” to:

Shifts in military demographics over the study period reflected increased proportions of female servicemembers and decreased proportions of non-Hispanic white servicemembers relative to other racial and ethnic groups [36].

The prior response comment about BMI values among heat illness subjects remaining steady comes from work in a separate study (part of the first author’s dissertation) and is not included in the discussion for this paper. 

Line 266-268 – No, there is not a specific term in our regression model for an overall cases trend. Rather, we applied block bootstrap resampling to control for universal long-term time-variable trends.

Line 275-276 – To clarify, the assessments were all based on quantitative values for each of the exposure indices and quantitative annual counts/rates of cases. However, from this dataset, we do not know the dates that cases occurred within a given year. The study design was constructed around the cross-sectional, annual-scale nature of the data, and the models remain valid for the heat-season metrics.

---

## [Decision Letter · Decision Letter 2]

25 Oct 2022

Heat stress illness outcomes and annual indices of outdoor heat at U.S. Army installations

PONE-D-22-02473R2

Dear Dr. Lewandowski,

We’re pleased to inform you that your manuscript has been judged scientifically suitable for publication and will be formally accepted for publication once it meets all outstanding technical requirements.

Kind regards,

Yanping Yuan

Academic Editor

PLOS ONE

Additional Editor Comments (optional):

Reviewers' comments:

Reviewer's Responses to Questions

**Comments to the Author**

1. If the authors have adequately addressed your comments raised in a previous round of review and you feel that this manuscript is now acceptable for publication, you may indicate that here to bypass the “Comments to the Author” section, enter your conflict of interest statement in the “Confidential to Editor” section, and submit your "Accept" recommendation.

Reviewer #2: All comments have been addressed

2. Is the manuscript technically sound, and do the data support the conclusions?

Reviewer #2: Yes

3. Has the statistical analysis been performed appropriately and rigorously? 

Reviewer #2: Yes

4. Have the authors made all data underlying the findings in their manuscript fully available?

Reviewer #2: Yes

5. Is the manuscript presented in an intelligible fashion and written in standard English?

Reviewer #2: Yes

6. Review Comments to the Author

Reviewer #2: thank you for addressing my concerns. I've done a bit more digging on my own time into the NLDAS datasets. thanks for your work and information you provided.

7. PLOS authors have the option to publish the peer review history of their article (what does this mean?). If published, this will include your full peer review and any attached files.

Reviewer #2: **Yes: **Peter J. Crank
